# Work extraction from quantum systems with bounded fluctuations in work

Jonathan G. Richens[1,2] & Lluis Masanes[2]

In the standard framework of thermodynamics, work is a random variable whose average is bounded by the change in free energy of the system. This average work is calculated without regard for the size of its fluctuations. Here we show that for some processes, such as reversible cooling, the fluctuations in work diverge. Realistic thermal machines may be unable to cope with arbitrarily large fluctuations. Hence, it is important to understand how thermodynamic efficiency rates are modified by bounding fluctuations. We quantify the work content and work of formation of arbitrary finite dimensional quantum states when the fluctuations in work are bounded by a given amount $c$. By varying $c$ we interpolate between the standard and minimum free energies. We derive fundamental trade-offs between the magnitude of work and its fluctuations. As one application of these results, we derive the corrected Carnot efficiency of a qubit heat engine with bounded fluctuations.

[1] Controlled Quantum Dynamics Theory Group, Department of Physics, Imperial College London, London SW7 2AZ, UK. [2] Department of Physics and Astronomy, University College London, Gower Street, London WC1E 6BT, UK. Correspondence and requests for materials should be addressed to J.G.R. (email: jonathan.richens08@ic.ac.uk).

Historically, thermodynamics has been a theory of macroscopic systems comprising of many particles. As we venture away from the thermodynamic limit we must question the validity of established principles. Recently, the problem of extracting work from a microscopic quantum system has received much attention[1–5]. The standard free energy is used to calculate the maximal amount of average work that can be extracted from a system in thermal contact with an infinite heat bath. Generally the work extracted on each running of the protocol fluctuates, but in the thermodynamic limit the relative size of fluctuations in work vanishes. However, in the case of microscopic systems, and systems that are far from equilibrium, fluctuations in the work can no longer be ignored. It is of significant practical importance that we understand these fluctuations in order to describe the behaviour of small and fragile machines such as quantum heat engines comprising of just a few qubits[6–8]. Realistic thermal machines are designed to operate at specific energies with a certain tolerance to fluctuations. Taking into account this inevitable fragility requires a modified free energy that tells us the average work associated with a process when fluctuations in that work are constrained.

One approach to dealing with fluctuations is to simply not allow for them. This is the tactic employed by single-shot thermodynamics, a recently developed approach to quantum thermodynamics inspired by the field of single-shot information theory[1,4,5]. The single-shot (deterministic) work associated with a process is given by the difference in minimum-free energy between initial and final states[1,4], which is generally significantly smaller that the standard free energy difference. The work cost of forming a state from the Gibbs state is given by the max-free energy[1,4], which is generally significantly larger than the deterministic work that can be extracted from the state. This discrepancy between the work cost and work content of states in the single-shot regime results in thermodynamic irreversibility when transforming between states. Furthermore, the set of allowed thermodynamic transformations in the single-shot regime are severely restricted. In this regime, it is possible for a state to undergo a transition $\rho \rightarrow \rho'$ deterministically (and without supplying work) on the condition that an infinite family of 'second laws' are satisfied[9]. Some transitions $\rho \leftrightarrow \rho'$ can only be achieved by supplying work in both forward and backward directions, resulting in a partial order on the set of states with respect to the resource of work[9]. This is in stark contrast to when we allow work to fluctuate freely, whereby all states can be inter-converted in a thermodynamically reversible manner.

To date the majority of thermodynamic protocols treat work either as an unconstrained random variable or a totally constrained (deterministic) quantity. In this article we explore the landscape of protocols that exist between these two regimes. We find that in many protocols, for example thermodynamically reversible cooling, the work must have fluctuations that diverge in size. This makes realising these protocols practically infeasible, especially for small or fragile machines. To this end we define the c-bounded work, giving the optimal average work $\langle w \rangle$ that can be achieved by any protocol when fluctuations of the random variable $w$ are bounded as

$$|w - \langle w \rangle| \leq c \qquad (1)$$

where $c$ is a adjustable parameter. In this article we explore how bounding work fluctuations in this way affects work extraction, state formation and the allowed state transformations of individual systems. We derive expressions for the c-bounded work that interpolate between these two regimes of deterministic and freely fluctuating work. We then apply these results to the study of a single qubit thermal engine, and derive a corrected Carnot efficiency when fluctuations in the work produced by the engine are constrained.

## Results

**The framework.** In this section we provide a precise description of our framework, describing the system, bath, work system and the set of allowed operations Figs 1–3.

We make use of the widely applied set-up for thermodynamic protocols of system, infinite thermal bath and a weight, which acts as a store and source of the work produced or consumed by a protocol[2,3,10]. In the following we set the Boltzmann constant $k_B$ to 1. The bath has infinite volume and it is in the Gibbs state $\rho_B = \frac{1}{\mathcal{Z}_B} e^{-\beta H_B}$, where $\beta$ is the inverse temperature, $H_B$ the Hamiltonian and $\mathcal{Z}_B$ the partition function.

The work system is modelled as a suspended weight with a continuous energy spectrum and Hamiltonian dependent only on its displacement $H_W = \int_\mathbb{R} dx\, x |x\rangle\langle x|$, where the orthonormal basis $\{|x\rangle, \forall x \in \mathbb{R}\}$ represents the position of the weight. To define work as a classical random variable $w$, the position of the weight is measured at the beginning and end of the protocol.

The system being transformed has Hilbert space of dimension $d$, initial state and Hamiltonian $(\rho, H_S)$, and final state and Hamiltonian $(\rho', H_S')$ (which may have no relation to the initial Hamiltonian, see Supplementary Note 1). It is useful to define the initial and final dephased states and their spectral decompositions

$$\lim_{T \to \infty} \int_0^T dt\, e^{-iH_S t} \rho\, e^{iH_S t} = \sum_s x_s |s\rangle\langle s|, \qquad (2)$$

$$\lim_{T \to \infty} \int_0^T dt\, e^{-iH_S' t} \rho'\, e^{iH_S' t} = \sum_s x_{s'} |s'\rangle\langle s'|. \qquad (3)$$

The two bases $|s\rangle$ and $|s'\rangle$ defined above, allow to write the spectral decompositions $H_S = \sum_s \mathcal{E}_s |s\rangle\langle s|$ and $H_S' = \sum_{s'} \mathcal{E}_{s'} |s'\rangle\langle s'|$. (Note that we use notation $x_{s'}$ and $\mathcal{E}_{s'}$ instead of $x_{s'}'$ and $\mathcal{E}_{s'}'$.) Finally, we assume that initially the joint state of system, bath and

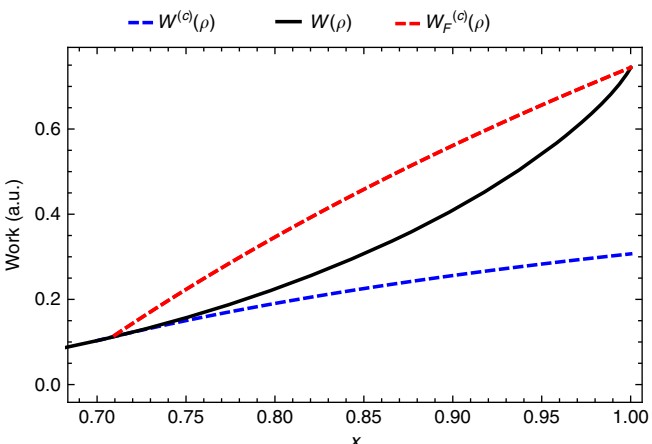

**Figure 1 | Variation of c-bounded work extraction and state formation further from equilibrium.** Figure shows the unbounded work $W(\rho)$, the c-bounded work content $W^{(c)}(\rho)$ and the c-bounded work of formation $W_F^{(c)}(\rho)$ for the state $\rho = x|0\rangle\langle 0| + (1-x)|1\rangle\langle 1|$ with Hamiltonian $H_S = \mathcal{E}|0\rangle\langle 0|$ with $\beta = 1$, $\mathcal{E} = 0.1$ and $c = 0.7$. Note that despite choosing a c-bound that allows for fluctuations of the order of the maximal work that can be extracted from the pure state $x = 1$, in general we can extract much less that this amount. There is a discontinuity in $W^{(c)}(\rho)$ at $x = 0$ where we recover $W^{(c)}(\rho) = W^{(\infty)}(\rho)$. Notice also that closer to the thermal state the dissipation (difference between the $W^{(c)}(\rho)$ or $W_F^{(c)}(\rho)$ and $W^{(\infty)}(\rho)$) is greater for state formations than work extraction, and this reverses as the state moves further from the Gibbs state.

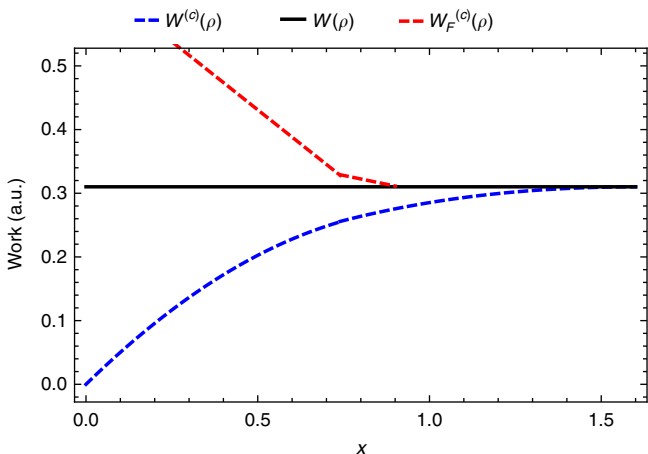

**Figure 2 | Variation of $c$-bounded work extraction and state formation further from equilibrium as a function of $c$ for fixed states.** Figure shows $W_F^{(c)}(\rho)$ and $W^{(c)}(\rho)$ versus $c$ for the trit state $\rho = 0.7|0\rangle\langle0| + 0.2|1\rangle\langle1| + 0.1|2\rangle\langle2|$ with Hamiltonian $H_s = \mathcal{E}_1|0\rangle\langle0| + \mathcal{E}_2|1\rangle\langle1|$ with $\mathcal{E}_1 = 0.1$, $\mathcal{E}_2 = 0.2$. $\beta$ is set to 1. For small $c$ the dissipation $|W^{(c)}(\rho) - W^{(\infty)}(\rho)|$ for the formation protocol is greater than for the extraction protocol, and for large $c$ the relationship is inverted. Note that for $c > 0.9$ it possible to thermodynamically reversibly prepare state $\rho$ but not to thermodynamically reversibly extract work from it.

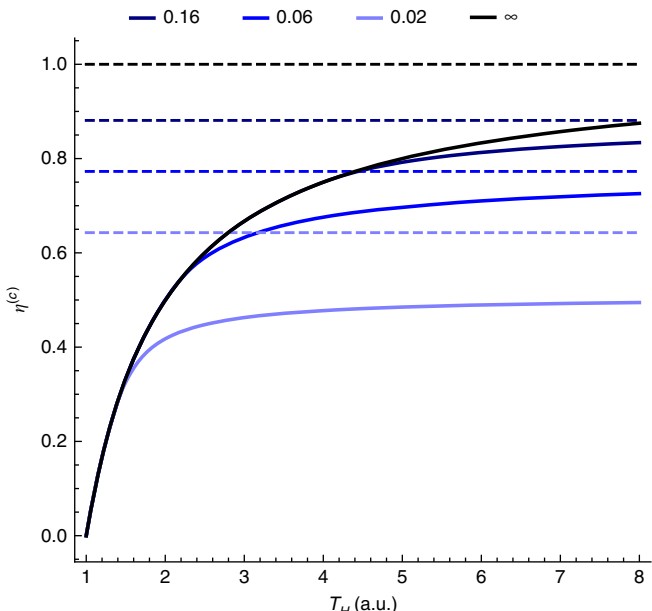

**Figure 3 | The corrected Carnot efficiency for various values of $c$ for a single qubit heat engine.** Figure shows the $c$-bounded Carnot efficiencies versus $T_H$ for single qubit engine with gap $\mathcal{E} = 0.1$ and $T_C = 1$. The black line gives the unbounded Carnot efficiency. The Blue lines give the $c$-bounded efficiencies with there corresponding $c$'s marked on the figure. The dashed lines give the maximum attainable efficiency in the limit of asymptotic temperature difference between $\beta_C/\beta_H \to \infty$.

weight is product $\rho \otimes \rho_B \otimes \rho_W$. We consider any process that is a joint transformation of system, bath and weight represented by a Completely Positive Trace Preserving (CPTP) map $\Gamma_{SBW}$ satisfying the following conditions:

Microscopic reversibility (Second Law): It has an (CPTP) inverse $\Gamma_{SBW}^{-1}$, which implies unitarity $\Gamma_{SBW}(\rho_{SBW}) = U\rho_{SBW}U^\dagger$.

Energy conservation (First Law): $[U, H_S + H_B + H_W] = 0$.

Independence from the 'position' of the weight: The unitary commutes with the translations on the weight $[U, \Delta_W] = 0$. The generator of the translations $\Delta_W$ is canonically conjugated to the position of the weight $[H_W, \Delta_W] = i$.

Classicality of work: Before and after applying the global map $\Gamma_{SBW}$ the position of the weight is measured, obtaining outcomes $|x\rangle$ and $|x+w\rangle$ respectively. The joint transformation of the system and work random variable $w$ is given by the map

$$\Lambda(\rho, w) = \int_{\mathbb{R}} dx \, \mathrm{tr}_{BW}\left[Q_{x+w} U(\rho \otimes \rho_B \otimes Q_x\rho_W Q_x)U^\dagger\right],$$
(4)

where $Q_x = |x\rangle\langle x|$ is a weight position projector.

The assumption that the dynamics of a closed system is reversible and conserves energy is widely used, because it corresponds to a common physical setup. The third condition implies that the reduced map on the system and bath is a mixture of unitaries, and therefore cannot decrease the entropy of the joint state of system and bath (See Result 1 in ref. 10). This ensures that the weight cannot be used as a source of non-equilibrium, and can be viewed as a necessary condition for defining work[2,11]. As a consequence of the fourth condition (classicality of work), the optimal work that can be extracted is determined by the dephased states of the system, hence the presence of coherences in the system cannot increase or decrease this amount (Supplementary Note 3). In the case of state formation, a state with coherences cannot be formed from a thermal state. However, for general state transformations (which we do not analyse in this paper) the presence of coherences in the initial and final states of system generate further constraints on

the work[12,13]. Regarding quantum definitions of work, several attempts have been made[14–16], but there is still no consensus on these definitions and their treatment of fluctuations. The problem of defining a truly quantum definition of work, or if such a definition exists, remains an important and open question[16–18].

**Deterministic work.** The single-shot work content of a system is given by the difference in minimum free energy $F^{\min}(\rho) = -\beta^{-1}\log\sum_s x_s^0 e^{-\beta\mathcal{E}_s}$ between the state $\rho$ and the thermal state

$$W^{(0)}(\rho) = \frac{1}{\beta}\log\mathcal{Z} - \frac{1}{\beta}\log\sum_s x_s^0 e^{-\beta\mathcal{E}_s},$$
(5)

where $x^0$ returns 1 if $x > 0$ and 0 if $x = 0$, and $\mathcal{Z} = \mathrm{tr}(e^{-\beta H_S})$ is the partition function of the system[1,4]. If $x_s$ has full rank then minimum free energy is $-\beta^{-1}\log\mathcal{Z}$. Therefore non-zero deterministic work can only be extracted from states that are not of full rank. The single-shot work of formation is

$$W_F^{(0)}(\rho) = \frac{1}{\beta}\log\mathcal{Z} + \frac{1}{\beta}\log\max_s x_s e^{\beta\mathcal{E}_s}.$$
(6)

In general we find that $W_F^{(0)}(\rho) > W^{(0)}(\rho)$, that is, it is not possible to form most states in a thermodynamically reversible manner. When a weight is not present, the necessary and sufficient condition for a state transformation $(\rho, H_S) \to (\rho', H_S')$ to be possible is given by the thermo-majorisation criteria[1]. If in addition a catalyst is used, the necessary and sufficient conditions are given in ref. 9 (for states that are diagonal in the energy eigenbasis). The key phenomenon is that in single-shot thermodynamics there is a partial order on states, that is, there are state transitions that are impossible, in both forward and backward directions, without supplying work. In contrast thermodynamic irreversibility and partial order are not observed

in the standard thermodynamic formalism, in which work is allowed to fluctuate freely, as we discuss next.

**Work with unbounded fluctuations**. The maximum average work that can be extracted from a system with the assistance of a heat bath with inverse temperature $\beta$ is given by the difference in free energy between $\rho$ and the Gibbs state. The free energy is given by $F(\rho) = \langle \mathcal{E} \rangle - \beta^{-1} S(\rho)$, $\langle \mathcal{E} \rangle = \mathrm{tr}(\rho\, H_S) = \sum_s x_s \mathcal{E}_s$ is the internal energy of the state, $S(\rho) = -\sum_s x_s \log x_s$ is the entropy of the de-phased state. The Gibbs state, with energy level occupation probabilities $x_s = \mathcal{Z}^{-1} e^{-\beta \mathcal{E}_s}$, is the unique state given $H_S$ and $\beta$ with the lowest free energy (given by $-\beta^{-1} \log \mathcal{Z}$). Therefore the optimal average work that can be extracted from an out of equilibrium state is given by $W^{(\infty)} = \beta^{-1} \log \mathcal{Z} + F(\rho)$. In the reverse process of state formation the work cost is also given by the difference in free energy between initial and final states. In other words, if we do not bound fluctuations in the work it is always possible to realize all state transformations in a thermodynamically reversible way. This poses the question— what is the minimum amount we must allow work to fluctuate in order for a transition to be achievable with a thermodynamically reversible protocol?

*Theorem* 1. The thermodynamically reversible process achieving the transition $(\rho, H_S) \to (\rho', H_S')$ with minimal fluctuations in work has work values $w(s'|s) = \beta^{-1} \log(x_s e^{\beta \mathcal{E}_s} / x_{s'} e^{\beta \mathcal{E}_{s'}})$. Therefore there exists a thermodynamically reversible process achieving the transition $(\rho, H_S) \to (\rho', H_S')$ with fluctuations in work less than or equal to $c$ if

$$e^{\beta(\Delta F - c)} \le \frac{x_s e^{\beta \mathcal{E}_s}}{x_{s'} e^{\beta \mathcal{E}_{s'}}} \le e^{\beta(\Delta F + c)} \quad \forall s, s', \tag{7}$$

where $\Delta F = F(\rho) - F(\rho')$ is the change in the standard free energy. This becomes a necessary and sufficient condition if the initial and/or or final state is diagonal in the energy eigenbasis

*Proof*: For a detailed proof see Supplementary Note 2.

Note that for any finite $c$ there exist states such that (7) cannot be satisfied. These bounds have strong consequences for the minimal fluctuations that can be achieved with a thermodynamically reversible protocol. For example

$$\lim_{x_i \to 0} \log \frac{x_s e^{\beta \mathcal{E}_s}}{x_{s'} e^{\beta \mathcal{E}_{s'}}} = - \lim_{x_{s'} \to 0} \log \frac{x_s e^{\beta \mathcal{E}_s}}{x_{s'} e^{\beta \mathcal{E}_{s'}}} = -\infty \tag{8}$$

Therefore as an energy level occupation probability tends to zero the work fluctuation associated with transitioning to or from this energy level diverges, negatively for work extraction and positively for state formation, when performing a thermodynamically reversible protocol. In either case we require $c \to \infty$ in order to satisfy inequalities (7).

Result 1 tells us that the further from equilibrium the initial or final states are, the larger the work fluctuations will be in thermodynamically reversible protocols. Note that cooling a system close to its ground state is an example of transitioning from the thermal state to a far from equilibrium state. Similarly, if we want to extract work form a far from equilibrium state using a thermodynamically reversible transformation we encounter the same divergence in fluctuations. The fluctuations can diverge even if the average work remains small (for example, if the system is a qubit with trivial Hamiltonian then $W \le \beta^{-1} \log 2$). These divergences have been previously noted in the recent study of absolute irreversibility[19,20].

Previous discussions of the inadequacy of the standard free energy in the nano-regime have focused on the definitions of work[4,5]. Here we add another criticism, that using the standard free energy to describe work we necessarily requires set-ups that can tolerate arbitrary fluctuations, which diverge in size for

processes with initial or final states that are increasingly far from equilibrium.

**Work with bounded fluctuations**. Motivated by these observations we define the c-bounded work content $W^{(c)}(\rho)$ as the maximum average work that can be extracted from state $\rho$ with initial Hamiltonian $H_S$ and final Hamiltonian $H_S'$, when the fluctuations of the work are constrained by $c$, as in equation (1). This notion of c-bounded work, and a generalization to include a small probability of failure, was proposed in ref. 4 but not developed beyond its definition. Analogously, we define the c-bounded work of formation $W_F^{(c)}(\rho)$ as the minimal average work that is necessary to create a state $\rho$ with Hamiltonian $H_S'$ from the Gibbs state (with respect to initial Hamiltonian $H_S$), such that fluctuations in the work are bounded by $c$.

*Theorem* 2. The c-bounded work content $W^{(c)}(\rho)$ and work of formation $W_F^{(c)}(\rho)$ are given by

$$W^{(c)}(\rho) = \frac{1}{\beta} \log \mathcal{Z}' - \frac{1}{\beta} \log \sum_s x_s^0 e^{-\beta(\mathcal{E}_s - \theta_s^{(c)})} \tag{9}$$

$$W_F^{(c)}(\rho) = \frac{1}{X_u} \left[ \sum_{s \in \mathcal{X}_u} \frac{x_s}{\beta} \log\left(x_s e^{\beta \mathcal{E}_s'} \mathcal{Z}\right) + c(1 - X_u) \right] \tag{10}$$

*Proof*: See Supplementary Notes 3 and 4 respectively.

Example calculations of the c-bounded work content and work of formation are shown in Figs 1 and 2. First we describe the terms in (9). $\mathcal{Z}' = \sum_{s'} e^{-\beta \mathcal{E}_{s'}}$ is the partition function of the final Hamiltonian $H_S'$. The second term can be viewed as a generalization of the minimum free energy (5) that allows for fluctuations in work $\theta_s^{(c)}$. The only difference to the minimum free energy is the term $e^{\beta \theta_s^{(c)}}$ included in the summation. $\theta_s^{(c)}$ is the fluctuation of the work value from the c-bounded average $W^{(c)}(\rho)$ given that the system was initially in state $|s\rangle$. To find the fluctuations associated with the optimal c-bounded work extraction protocol we must partition the energy levels into three disjoint subsets $\{1, 2, \ldots, d\} = \mathcal{X}_u \cup \mathcal{X}_+ \cup \mathcal{X}_-$, representing the energy levels with positive $\mathcal{X}_+$, negative $\mathcal{X}_-$ and unbounded $\mathcal{X}_u$ fluctuations. We also define

$$X_u = \sum_{s \in \mathcal{X}_u} x_s, \tag{11}$$

$$X_\pm = \sum_{s \in \mathcal{X}_\pm} x_s. \tag{12}$$

The algorithm for determining the partition $\mathcal{X}_u \cup \mathcal{X}_+ \cup \mathcal{X}_-$, which requires the checking of at most $d - 1$ inequalities, is described Supplementary Note 3B. Once we have determined the partition, the fluctuations are given by

$$\theta_s^{(c)} = \begin{cases} \frac{1}{\beta} \log(x_s e^{\beta \mathcal{E}_s}) - v, & s \in \mathcal{X}_u \\ + c, & s \in \mathcal{X}_+ \\ - c, & s \in \mathcal{X}_- \end{cases} \tag{13}$$

where

$$v = \frac{1}{X_u} (F_u + c(X_+ - X_-)) \tag{14}$$

$$F_u(\rho) = \sum_{s \in \mathcal{X}_u} x_s \log(x_s e^{\beta \mathcal{E}_s}) \tag{15}$$

where $F_u(\rho)$ is the free energy calculated for the unbounded partition only. Note that $W^{(c)}(\rho)$ can be written in the more compact form

$$W^{(c)}(\rho) = \frac{1}{\beta} \log \mathcal{Z}' - \frac{1}{\beta} \log\left(X_u e^{-\beta v} + \mathcal{Z}_+ e^{\beta c} + \mathcal{Z}_- e^{-\beta c}\right)$$

$$\tag{16}$$

where

$$\mathcal{Z}_{\pm} = \sum_{s \in \mathcal{X}_{\pm}} e^{-\beta \mathcal{E}_s} \qquad (17)$$

are the partition functions calculated over the positive and negative bounded partitions respectively. In Supplementary Notes 3 and 4 we find that in the optimal work extraction and state formation protocols the final/initial state of the system is the Gibbs state, which is diagonal in the energy eigenbasis. Hence equations (9) and (10) give the optimal work for arbitrary quantum states (Supplementary Note 1). For a two-level quantum system $d = 2$, the work content can be expressed succinctly as

$$W^{(c)}(\rho) =$$
$$\begin{cases} \frac{1}{\beta}\log\mathcal{Z}' - \frac{1}{\beta}\log\left[e^{+\beta c}\left(1 + e^{-\beta(\mathcal{E} + c/x_1)}\right)\right] & \text{if } c < \xi \\ \frac{1}{\beta}\log\mathcal{Z}' - \frac{1}{\beta}\log\left[e^{-\beta c}\left(1 + e^{-\beta(\mathcal{E} - c/x_1)}\right)\right] & \text{if } c > -\xi \\ \frac{1}{\beta}\log\mathcal{Z}' + F(\rho) & \text{otherwise} \end{cases} \qquad (18)$$

where

$$\xi = \frac{1}{\beta}\ln(1 - x_1) - F(\rho). \qquad (19)$$

Without loss of generality we have assumed above $x_1 \geq x_2$ and we define $\mathcal{E} = \mathcal{E}_1 - \mathcal{E}_2$. The above expression derives from equation (16) and the partitioning algorithm given in Supplementary Note 3.

Returning to the $c$-bounded work of formation, equation (10) and the algorithm for finding the state space partition giving the $c$-bounded work of formation equation (10) is detailed in Supplementary Note 4. Note that the hamiltonian of $\rho$, the final state of the system, is given by $H'_S = \sum_j \mathcal{E}'_j |j\rangle\langle j|$. The work of formation for a two level system can be succinctly stated as

$$W_F^{(c)}(\rho) = \begin{cases} \frac{1}{\beta}\log\mathcal{Z} - F(\rho), & c \geq \xi \\ \frac{1}{\beta}\log\left(x\,e^{\beta\mathcal{E}'}\mathcal{Z}\right) + c\frac{1-x}{x}, & c < \xi \end{cases} \qquad (20)$$

where $\rho = x|0\rangle\langle 0| + (1-x)|1\rangle\langle 1|$, $x \geq 1/2$ and $H_S = \mathcal{E}|0\rangle\langle 0|$ is the Hamiltonian of the initial Gibbs state.

We now summarize some of the properties of the $c$-bounded work.

*Theorem* 3. The $c$-bounded work is related to the non-fluctuating work by inequalities

$$W^{(c)}(\rho) \leq W^{(0)}(\rho) + c \qquad (21)$$

$$W_F^{(c)}(\rho) \geq W_F^{(0)}(\rho) - c \qquad (22)$$

becoming strict inequalities for $c > 0$

*Proof*: See Supplementary Note 5.

These inequalities imply a fundamental trade-off between work and fluctuations. To do better than single-shot work extraction/state formation, our work must have fluctuations that are greater than the increase in work/decrease in work cost with respect to the deterministic work. In Supplementary Note 5 we show that the $c$-bounded work distributions that give $W^{(c)}(\rho)$ and $W_F^{(c)}(\rho)$ obey the Jarzinski equality[21]. In Supplementary Note 5 we prove that $\lim_{c \to 0} W^{(c)}(\rho) = W^{(0)}(\rho)$ and similarly for $W_F^{(c)}(\rho)$.

For the interested reader, it is simple to see that for any finite $c$ a partial ordering of the states w.r.t work emerges. A simple way to observe this is to choose a qubit state $\rho$ with Hamiltonian $H_S$ and a thermal qubit state $\gamma$ with Hamiltonian $H'_S \neq H_S$ such that neither state thermo-majorizes the other (see ref. 1 for examples). For any two such states there is a value of $c$ below which $W^{(c)}(\rho)$ (for $\rho \to \gamma$) is negative and $W_F^{(c)}(\rho)$ (for $\gamma \to \rho$) is positive, that is, it costs work to perform both the forward and backwards transitions. Note that there are states with the same standard free energy that exhibit a partial order for finite $c$. Allowing the weight to fluctuate allows us

to transition between these states freely. This is an example of how weight is not just a resource for extracting additional fluctuating work but in accommodating dynamics, even when on average its displacement remains zero. It is an interesting open question to determine how much we must allow work to fluctuate to allow a transition $\rho \to \rho'$ to be achieved without costing work.

**Qubit Carnot engine**. In this section we find the $c$-bounded Carnot efficiency for a qubit Carnot engine model, that is, the maximal efficiency the qubit engine can reach given that fluctuations in the work it produces are bounded by $c$. We use the same single qubit engine model as described in ref. 2. The engine operates by moving a qubit $\rho$ with Hamiltonian $H_s = \mathcal{E}|0\rangle\langle 0|$ between two baths of inverse temperature $\beta_H$ and $\beta_C$, with $\beta_H < \beta_C$. The qubit has state $\rho_{H,C} = \mathcal{Z}_{H,C}^{-1} e^{-\beta_{H,C}\mathcal{E}}|0\rangle\langle 0| + \mathcal{Z}_{H,C}^{-1}|1\rangle\langle 1|$ when in equilibrium with the hot/cold bath, where $\mathcal{Z}_{H,C} = 1 + e^{-\beta_{H,C}\mathcal{E}}$. The engine cycle begins with the qubit in thermal equilibrium with the cold bath. In the first half of the cycle it is then placed in contact with the hot bath and work is extracted. In the second step of the cycle the qubit is returned to the cold bath and work is extracted a second time. In Supplementary Note 6 we show that, in the case that fluctuations are not bounded, it is possible to reach Carnot efficiency with this engine, as shown in ref. 2.

$$\eta_{\text{Carnot}} = 1 - \frac{\beta_H}{\beta_C} \qquad (23)$$

In the case that $c$ is finite, the work extracted in the first half of the cycle is given by

$$W_1^{(c)}(\rho) = \begin{cases} \frac{1}{\beta_H}\log\left(\frac{\mathcal{Z}_H}{\mathcal{Z}_C}\right) + \text{tr}[H_s\rho_C]\left(\frac{\beta_H - \beta_C}{\beta_H}\right), & \text{if } A > c \\ \frac{1}{\beta_H}\log\left(\frac{\mathcal{Z}_H e^{\beta_H c}}{e^{\beta_H \mathcal{Z}_C} + e^{-\mathcal{E}\beta_H}}\right), & \text{if } A \leq c \end{cases} \qquad (24)$$

where

$$A = \frac{\mathcal{E}}{\mathcal{Z}_C}\left(\frac{\beta_C - \beta_H}{\beta_H}\right) \qquad (25)$$

if $A \leq c$ we simply extract the difference in free energy between the two thermal states, otherwise we extract the $c$-bounded work of $\rho_C$ in contact with the bath $\beta_H$. Similarly, on the second part of the cycle we extract

$$W_2^{(c)}(\rho) = \begin{cases} \frac{1}{\beta_C}\log\left(\frac{\mathcal{Z}_C}{\mathcal{Z}_H}\right) + \text{tr}[H_s\rho_H]\left(\frac{\beta_C - \beta_H}{\beta_H}\right), & \text{if } B > c \\ \frac{1}{\beta_C}\log\left(\frac{\mathcal{Z}_C e^{-\beta_C c}}{e^{-c\beta_C}\mathcal{Z}_H + e^{-\mathcal{E}\beta_C}}\right), & \text{if } B \leq c \end{cases} \qquad (26)$$

where

$$B = \frac{\mathcal{E}}{\mathcal{Z}_H}\left(\frac{\beta_C - \beta_H}{\beta_C}\right) \qquad (27)$$

Note that satisfying (27) implies that (25) is also satisfied, therefore breaking inequality (25) is the condition for achieving Carnot efficiency in this model. Also note that $B$ gives the minimum worst case fluctuation of the work extracted by this engine when operating thermodynamically reversibly. The efficiency is given by the ratio of the heat flow from the hot bath to the total work extracted in the cycle. The heat flow from the hot bath is found by applying the 1$^{\text{st}}$ law of thermodynamics, $Q_H = \Delta\langle\mathcal{E}\rangle(\rho_H \to \rho_C) + W_1^{(c)}$ where $\Delta\langle\mathcal{E}\rangle(\rho_H \to \rho_C)$ is the change in the systems internal energy in the first part of the cycle. Therefore the $c$-bounded efficiency of the engine is given by

$$\eta^{(c)} = \frac{W_1^{(c)} + W_2^{(c)}}{\Delta U + W_1^{(c)}} \qquad (28)$$

In the case that $c \to \infty$, we recover the Carnot efficiency, which is bounded from above by 1, that is, we recover unit efficiency in the limit that $\beta_C/\beta_H \to \infty$. For any finite $c$ this is no longer the case, with the maximal efficiency give by

$$\eta_{\max}^{(c)} = \lim_{\beta_C/\beta_H \to \infty} \eta^{(c)} = 1 - \frac{\mathcal{E}}{2(2c + \mathcal{E})} \qquad (29)$$

This gives an upper limit on the efficiency of the single qubit engine protocol described above, which is dependent only on the Hamiltonian of the qubit and the parameter $c$. The corrected Carnot efficiencies for various values of $c$ are shown in Fig. 3.

As $\mathcal{E} \to 0$, $\eta_{\max}^{(c)} \to 1$, but the work extracted tends to zero as the Gibbs states associated with the two bath temperatures become indistinguishable. For $c \to 0$ we get that $\eta_{\max}^{(c)}$ is bounded from below by 1/2, but at $c = 0$ no work can be extracted as the thermal states are of full rank, giving $\eta_{\max}^{(0)} = 0$. Therefore we find that, although this engine cannot run at non-zero efficiency in the single-shot regime, if we allow for arbitrarily small fluctuations it is possible in principle to reach a maximum efficiency greater that 1/2 (although the fluctuations in work will still be of the order of the work extracted, given inequality (21)). Similar results relating to the single-shot regime are discussed in ref. 22.

## Discussion

In this article we have derived tight bounds on the minimal fluctuations in work associated with thermodynamically reversible protocols, for which the average work is given by difference in free energy between initial and final states. We have found that thermodynamically reversible protocols have fluctuations that diverge in size as the relative athermality of initial or final states increases.

Motivated by this we have presented a framework for computing the work associated with a thermodynamic process under arbitrary convex bounds. We have derived the $c$-bounded work content and work of formation of arbitrary quantum states, which can be understood as modified free energies that interpolate between the standard and single-shot free energies. By exploring this new territory, we have found that the phenomenology of single-shot thermodynamics, namely thermodynamic irreversibility and a partial order of states with respect to work, are to some extent present for any finite $c$. Furthermore we have found that it is impossible to extract more that the deterministic work content of a system without necessitating fluctuations that are greater than the gain in work (and similarly for the work cost of state formation). One potential avenue for extending these results would be to consider a more general definition of $c$-bounded work that includes a small probability of failure in the work extraction process (as proposed in ref. 4)

An interesting open question is to what extent we must allow work to fluctuate in order to allow for a given state transformation. Answering this question would require the extension of the results presented in this article to processes with arbitrary initial and final states, including the case where both initial and final states contain coherences between energy levels. In Supplementary Note 1C we show that, under the assumption that the protocol is independent on the position state of the weight, the 'coherence modes'[12,13,23] evolve independently under the action of the thermal map. This lays the ground for future investigations into how the presence of coherences affects the allowed thermal operations in the case that work is allowed to fluctuate.

Finally, we have used the $c$-bounded work to study how bounding work fluctuations affects the efficiency of a single qubit nano-engine, and have derived an upper bound on efficiency of this engine that depends only on $c$ and the engine's Hamiltonian, establishing a fundamental trade-off between a the engines efficiency and the fluctuations in the work it produces. This opens the door to correcting the efficiency for general thermodynamic protocols, taking into account the fragility of realistic machines that cannot tolerate large fluctuations in work.

Given that there are many thermal engine models that can reach Carnot efficiency in the case the fluctuations in work are unbounded, it would be of interest to determine the optimal engine with respect to minimizing fluctuations in work whilst maximizing efficiency or the power produced. Furthermore, it is well known that in the thermodynamic limit the relative size of fluctuations in work to the average tends to zero. It would be of interest to determine if it is possible to design engines operating far from the thermodynamic limit that achieve a similar quasi-deterministic work output with non-zero power. For example it could be possible, through clever choice of the working system Hamiltonian, or by controlling interactions between a small number of systems that constitute the working system, to find engine models that achieve quasi-deterministic work output without needing to take the thermodynamic limit. Further work in this direction would provide invaluable insights for designing realistic nano-engines that are robust to fluctuations in work.

## Methods

**Proofs and derivations.** The proofs are contained in Supplementary Information. In Supplementary Note 1 we address the preliminaries and framework within which we derive our proofs, including the framework of thermal operations with fluctuating work, changing Hamiltonians, coherences and reducing quantum the protocols to classical protocols, and proofs that our framework is both general and optimal. In Supplementary Note 2 we derive the form of the work optimal protocols with unbounded fluctuations, and derive Result 1. In Supplementary Note 3 we derive the $c$-bounded work content, and the corresponding state partitioning algorithm. In Supplementary Note 4 we derive the $c$-bounded work of formation, and the corresponding state partitioning algorithm. In Supplementary Note 5 we show that the standard and minimum free energy can be recovered in the limits $c \to \infty$ and $c \to 0$ respectively, and derive the trade-off bounds relating the optimal work to the size of the worst-case fluctuations. In Supplementary Note 6 we derive the $c$-bounded Carnot for the single qubit heat engine model.

**Data availability.** Data sharing is not applicable to this article, as no data sets were generated or analysed during this study.

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

## Acknowledgements

J.R. would like to thank Jochen Gemmer, Mart Perarnau-Llobet, Alvaro Alhabra and Chris Perry for discussions. J.R. is supported by ESPRC.

## Author contributions

Both authors contributed equally to this work

## Additional information

**Competing financial interests:** The authors declare no competing financial interests.

**Publisher's note**: 

