## [Peer Review File · Nature Communications]

Reviewer #1 (Remarks to the Author):

The authors introduce a result that interpolates between two notions of thermodynamics work in quantum realm. They derive several tight bounds smoothly relating the relevant work definitions via a bounding parameter c .

The work is an original contribution that has the potential to become part of a future standard framework on a topic that has certainly sparked a certain amount of discussions in the community [1,2,3] (Ref 2 is also cited by the authors). This contribution is particularly relevant as it operationally manages to overcome two of the main objections to quantum mechanical definitions of work, that were:

(i) Deterministic work is virtually impossible to achieve, as it would require preparing quantum systems without some level of noise. And it is disconnected from the practical design of quantum machines, as there are undeniably useful tasks that may withstand some modest amount of fluctuating energies.

(ii) average work (i.e. free energy differences using von Neumann entropies). While they deliver quite ultimate bounds on achievable transformations, they do so only in a very static picture. E.g. in practice not all infinitely many potential eigenstates that are accessed will be stable. And it may not be desirable to have the energies fluctuate in an unbounded way (If I want my machine to deliver some energy to a certain region, it may just be unpractical to have it potentially deliver orders of magnitude more/less with a small probability).

Fluctuation bounded work is a great idea to overcome both criticisms, by interpolating between these two regimes. The bounding parameter c thus becomes an operationally relevant boundary condition as the amount of fluctuation that is tolerated for the specific operational task.

As such the paper is in my opinion a key contribution to the rapidly developing field of quantum thermodynamics and will certainly have a significant amount of impact in the community, especially since it has the potential to unite approaches rooted in quantum information.

The only downside is that the authors only consider a very classical notion of work in all cases (i.e. the prior and posterior dephasing makes all involved states essentially classical). I would find it even more exciting if a similar concept would emerge for genuinely quantum contributions to work, as obviously also off-diagonal values can have an important contribution to the energetics of quantum system (that can however only be accessed with some level of coherence/timing). It would, given the great technical contribution and clarity of writing, be unfair to hold this against the current manuscript.

In summary I can strongly recommend publication of this manuscript in Nature communications.

[1] Phys. Rev. E 93 022131 (2016)

[2] arXiv:1504.05056

[3] arXiv:1606.08368

Reviewer #2 (Remarks to the Author):

Recent years have seen considerable interest and progress in the topic of extending thermodynamics so that it applies at the level of individual quantum systems. Some of the basic questions which one can ask in this setting concern which state transformations are possible in a thermodynamically reversible manner? How much work does it take to produce a given state starting from a thermal state? How much work can be extracted in the process of bringing a quantum system to thermal equilibrium with a bath? There are many subtle aspects to these questions, and more generally when studying quantum thermodynamics, and so it has been

important to take a varied approach to these questions to gain a rounded answer.

In this manuscript, the authors make an important and substantial contribution to this line of research. At this level, it is intuitively clear that work in general will be a fluctuating quantity, due in part to the probabilistic nature of quantum theory itself. The authors propose here to study the above questions, concerning how much work can be extracted, or how much must be spent, by the best possible 'protocol', with the additional constraint that the fluctuations in work must be bounded, by some arbitrary amount. This is indeed a natural restriction to place on protocols, since one does not intuitively expect systems to be able to cope with arbitrarily large fluctuations in the energy they accept or must provide. Understanding how the optimal protocols depend on the bound of fluctuations therefore provides an important insight into the nature of quantum thermodynamics for individual systems.

The authors obtain a number of key results. This first result shows that in order to achieve the optimal work extraction from a quantum system, then in fact the fluctuations are necessarily arbitrarily large. More precisely, they show that given an upper bound on the fluctuations allowed, one can always find a state (which will be very far from equilibrium) which requires larger fluctuations in order to carry out the optimal protocol which will extract the optimal amount of work (and similarly for creating the state). In essence, this shows the necessity in a concrete manner for studying fluctuation-bounded protocols.

They then manage to precisely characterise the amount of work extractable from a system, and the amount of work necessarily spent to create a system, with a bounded-fluctuation protocol. They furthermore show that there is a precise form of trade-off between the fluctuations and the work.

Finally, they use their framework to study a simple model of a heat engine (that runs between two thermal reservoirs at differing temperatures), and show that the efficiency of such a machine is also bounded as a function of the fluctuations.

I find these results to be extremely interesting, and believe they make an important contribution to our understanding of quantum thermodynamics. My recommendation is that this manuscript should be accepted for publication. Notwithstanding, I have a number of questions and comments below, which I believe the authors should address (where appropriate) before the paper can be accepted.

1. I think the authors should comment on the robustness of their definition. The concern that the definition strictly allows for no fluctuations above the amount specified, and that this might in fact be too stringent a requirement. One could also consider, like the in the deterministic-work setting, allowing for an amount ϵ of fluctuation above c . This would essentially play the role of smoothing in the former context. In this case, it seems to me that the examples given in Result 1 may no longer hold, since I believe there the fluctuations indeed decrease rapidly (i.e. these are tail effects), which could be accounted for. I fully understand the choice of the authors for choosing the strict bound in the first instance, but I believe it would be interesting and important to consider the case of smoothing, if only in further work. this happens.

2. Related to the above, could the authors also comment on the relation of their definition to that of Aberg in [4], of ϵ , δ -work extraction.

3. Is there any scope to relax the constraint of the work system being measured at the beginning and the end of the protocol? I see that this essentially makes the problem classical, so that one can talk about random variables, but full generality one could consider situations where this doesn't happen (i.e. the weight might be used at the end of the protocol in another quantum thermodynamic protocol, via a controlled unitary, such that one can never think of a final perfect projective measurement taking place). Also, one might also consider states like coherent states to

be other natural classes of states for the weight. It would be interesting to know what can be said, if anything, in such cases.

4. Do the authors consider arbitrary change of system Hamiltonian $H_s \rightarrow H_{s'}$, or only ones where the eigenvalues change $E_s \rightarrow E_{s'}$, but the eigenstates remain constant?

5. The authors have chosen not to present any of the proofs of the key results in the main text. This I can understand, for reasons of space. What might be helpful for the interested reader is a sentence or two, or a short paragraph, outlining the proof techniques used, or giving some possible intuitive explanation behind the proof strategy. I believe this would make the paper more accessible. As an example, for result 1, does the necessity of unbounded fluctuations essentially come from the fact that for a thermodynamically reversible protocol one can only really interact subsystems whose states are almost identical (up to normalisation). Therefore, in a system where one needs to make a transition between system states that have populations ratios that diverge, one necessarily needs to use a subsystem from the bath with the same property, which (for fixed temperature), amounts to taking levels whose energy diverges, hence leading to unbounded fluctuations?

6. I believe there are some inconsistencies between the notation in the main text and the appendices, and some notation that is used in the main text that is not defined before use. As an example, $w(s|s')$ I believe is undefined in the main text, and in the appendix is $w_{\{s,s'\}}$? I would ask the authors to carefully check for consistency, again to help with the accessibility of the manuscript.

7. A number of small typos: I think in (2) - (3), the left hand side needs to be normalised (i.e. this is an average over time?); in (4) the tensor product symbols are missing, also in the appendix in places; [9] is only necessary and sufficient for diagonal states (the text could be misread to mean that it is sufficient in general); There is a change from $S(\rho)$ to $S(x_s)$ when it is introduced;

Response to the reviews - NCOMMS-16-14043

Dear Federico,

We are glad to know that the two reviews are very positive, and that both recommend the manuscript for publication. We thank them for their comments and suggestions, which we address in what follows.

Reviewer 1 says that this paper is a “key contribution to the rapidly developing field of quantum thermodynamics and will certainly have a significant amount of impact”, and that it “has the potential to become part of a future standard framework”. The reviewer suggests that the work could be further improved by including an analysis of fluctuations with a fully quantum notion of work, although they add that “It would, given the great technical contribution and clarity of writing, be unfair to hold this against the current manuscript”.

Reviewer 1: “The only downside is that the authors only consider a very classical notion of work in all cases (i.e. the prior and posterior dephasing makes all involved states essentially classical). I would find it even more exciting if a similar concept would emerge for genuinely quantum contributions to work, as obviously also off-diagonal values can have an important contribution to the energetics of quantum system (that can however only be accessed with some level of coherence/timing). It would, given the great technical contribution and clarity of writing, be unfair to hold this against the current manuscript.”

We agree that these issues raised by the reviewer are important, however there is still no consensus on how to define work fluctuations in a fully quantum setup, and it is safe to say that defining a truly quantum definition of work is a significant and important open question. This can be seen in the fact that most quantum versions of “fluctuation theorems”, like Jarzynski’s Equality, use the same (classical) work definition than us. Some attempts beyond this are detailed in [1, 2, 3, 4, 5], and our future research will follow this direction. This notwithstanding, we agree that it will improve the quality of the paper to acknowledge these issues more clearly and allude to attempts to go beyond the definition of work that we make use of. We have re-written paragraph 6 of the Results section to address these issues

and cite the relevant existing papers.

Reviewer 2

point 1: *“I think the authors should comment on the robustness of their definition. The concern that the definition strictly allows for no fluctuations above the amount specified, and that this might in fact be too stringent a requirement. One could also consider, like the in the deterministic-work setting, allowing for an amount ϵ of fluctuation above c . This would essentially play the role of smoothing in the former context. In this case, it seems to me that the examples given in Result 1 may no longer hold, since I believe there the fluctuations indeed decrease rapidly (i.e. these are tail effects), which could be accounted for. I fully understand the choice of the authors for choosing the strict bound in the first instance, but I believe it would be interesting and important to consider the case of smoothing, if only in further work. this happens.”*

We agree with the referee that it would be of interest to consider more general definitions of the c -bounded work, such as including probability of failure in the work extraction protocol (as proposed by Aberg in [6]). We have included a discussion of this point in the third paragraph of the introduction and the first paragraph of the discussion.

point 2: *“Related to the above, could the authors also comment on the relation of their definition to that of Aberg in [4], of ϵ, δ -work extraction.”*

As the referee points out, the above definition of smoothed c -bounded work was proposed by Aberg in [6], but not developed beyond its definition. We have commented on this in the first sentence after Equation (1) in the third paragraph of the introduction.

point 3: *“Is there any scope to relax the constraint of the work system being measured at the beginning and the end of the protocol? I see that this essentially makes the problem classical, so that one can talk about random variables, but full generality one could consider situations where this doesn't happen (i.e. the weight might be used at the end of the protocol in another quantum thermodynamic protocol, via a controlled unitary, such that one can never think of a final perfect projective measurement taking place). Also, one*

might also consider states like coherent states to be other natural classes of states for the weight. It would be interesting to know what can be said, if anything, in such cases.”

This is the same question risen by Reviewer 1. The answer is that there is still no consensus on how to define work fluctuations in a fully quantum setup. We are currently researching in this direction. On page 2, we clarify the role of coherences in our scheme (referee 1) , and refer to other works addressing definitions of quantum work

point 4: *“Do the authors consider arbitrary change of system Hamiltonian $H_s \rightarrow H'_s$, or only ones where the eigenvalues change $E_s \rightarrow E'_s$, but the eigenstates remain constant?”*

We consider arbitrary changes of the System’s Hamiltonian, as detailed in Supplementary note 1B. On page 2 we have included brief note clarifying that we achieve arbitrary Hamiltonian changes

point 5: *“The authors have chosen not to present any of the proofs of the key results in the main text. This I can understand, for reasons of space. What might be helpful for the interested reader is a sentence or two, or a short paragraph, outlining the proof techniques used, or giving some possible intuitive explanation behind the proof strategy. I believe this would make the paper more accessible. As an example, for result 1, does the necessity of unbounded fluctuations essentially come from the fact that for a thermodynamically reversible protocol one can only really interact subsystems whose states are almost identical (up to normalisation). Therefore, in a system where one needs to make a transition between system states that have populations ratios that diverge, one necessarily needs to use a subsystem from the bath with the same property, which (for fixed temperature), amounts to taking levels whose energy diverges, hence leading to unbounded fluctuations? ”*

We have included in the second paragraph after result 1 (starting “As an example to why...”) an example to illustrate why we can expect that these divergent fluctuations occur in thermodynamically reversible transformations, using the standard language of adiabats and isotherms. Furthermore we have included a brief discussion of the proof techniques in Result 1 on

page 3 of the paper, as per the referees suggestion.

point 6: *“I believe there are some inconsistencies between the notation in the main text and the appendices, and some notation that is used in the main text that is not defined before use. As an example, $w(s|s')$ I believe is undefined in the main text, and in the appendix is $w_{s,s'}$? I would ask the authors to carefully check for consistency, again to help with the accessibility of the manuscript.”*

We have corrected these notational inconsistencies.

point 7: *“A number of small typos: I think in (2) - (3), the left hand side needs to be normalised (i.e. this is an average over time?); in (4) the tensor product symbols are missing, also in the appendix in places; [9] is only necessary and sufficient for diagonal states (the text could be misread to mean that it is sufficient in general); There is a change from $S(\rho)$ to $S(x_s)$ when it is introduced; ”*

Typos corrected. Normalisation corrected on equations (2)-(3)

Furthermore, we have included changes to the article to comply with all requirements for publication in this journal, including

- author email added
- abstract shortened to obey the 150 word limit
- changed section headings to befit the guidelines
- changed supplementary section format and references to befit the guidelines
- included methods section and changed sections / headings
- included statement of data availability, acknowledgements, author contributions and competing financial interests
- supplementary information put in separate file and supplementary bibliography included
- new citations added

Best regards,

Jonathan Richens (on behalf of the authors)

References

- [1] Johan Aberg. Fully quantum fluctuation theorems. *arXiv preprint arXiv:1601.01302*, 2016.
- [2] Álvaro M Alhambra, Lluís Masanes, Jonathan Oppenheim, and Christopher Perry. The second law of quantum thermodynamics as an equality. *arXiv preprint arXiv:1601.05799*, 2016.
- [3] Marti Perarnau-Llobet, Elisa Baumer, Karen V Hovhannisyan, Marcus Huber, and Antonio Acín. Quantum fluctuations of work and generalised quantum measurements. *arXiv preprint arXiv:1606.08368*, 2016.
- [4] R Gallego, J Eisert, and H Wilming. Thermodynamic work from operational principles. *arXiv preprint arXiv:1504.05056*, 2015.
- [5] Peter Talkner and Peter Hanggi. Aspects of quantum work. *Physical Review E*, 93(2):022131, 2016.
- [6] Johan Åberg. Truly work-like work extraction via a single-shot analysis. *Nature communications*, 4, 2013.

Reviewer #2 (Remarks to the Author):

I am happy with the author's replies to my questions and comments in the previous round of review. I am furthermore happy with the changes that have been made to the manuscript in light of my comments and suggestions, as well as those made in response to the other reviewer.

I therefore reiterate my previous suggestion, that this manuscript makes an important contribution to the literature on quantum thermodynamics, and recommend it for publication in Nature Communications.

Reply to referees: Work extraction from quantum systems with bounded fluctuations in work

Jonathan G. Richens^{1,2} and Lluís Masanes²

¹*Controlled Quantum Dynamics theory group, Department of Physics,
Imperial College London, London SW7 2AZ, UK.*

²*Department of Physics and Astronomy, University College London, Gower Street, London WC1E 6BT, UK.*

Dear Dr. Levi,

We are delighted to be accepted for publication. Please find in this letter all requested responses.

Referee 2) “I am happy with the author’s replies to my questions and comments in the previous round of review. I am furthermore happy with the changes that have been made to the manuscript in light of my comments and suggestions, as well as those made in response to the other reviewer.

I therefore reiterate my previous suggestion, that this manuscript makes an important contribution to the literature on quantum thermodynamics, and recommend it for publication in Nature Communications.”

The referee request no further alterations and recommends us for publication. As such, no replies are given.

Best regards,

Jonathan Richens